# Spin-orbital Jahn-Teller bipolarons

Lorenzo Celiberti [1,2], Dario Fiore Mosca [1,3,4], Giuseppe Allodi[5], Leonid V. Pourovskii [3,4], Anna Tassetti[2], Paola Caterina Forino[2], Rong Cong[6], Erick Garcia[6], Phuong M. Tran[7], Roberto De Renzi [5], Patrick M. Woodward[7], Vesna F. Mitrović[6], Samuele Sanna[2] & Cesare Franchini [1,2] ✉

Polarons and spin-orbit (SO) coupling are distinct quantum effects that play a critical role in charge transport and spin-orbitronics. Polarons originate from strong electron-phonon interaction and are ubiquitous in polarizable materials featuring electron localization, in particular 3d transition metal oxides (TMOs). On the other hand, the relativistic coupling between the spin and orbital angular momentum is notable in lattices with heavy atoms and develops in 5d TMOs, where electrons are spatially delocalized. Here we combine ab initio calculations and magnetic measurements to show that these two seemingly mutually exclusive interactions are entangled in the electron-doped SO-coupled Mott insulator $Ba_2Na_{1-x}Ca_xOsO_6$ ($0 < x < 1$), unveiling the formation of *spin-orbital bipolarons*. Polaron charge trapping, favoured by the Jahn-Teller lattice activity, converts the Os $5d^1$ spin-orbital $J_{eff} = 3/2$ levels, characteristic of the parent compound $Ba_2NaOsO_6$ (BNOO), into a bipolaron $5d^2$ $J_{eff} = 2$ manifold, leading to the coexistence of different J-effective states in a single-phase material. The gradual increase of bipolarons with increasing doping creates robust in-gap states that prevents the transition to a metal phase even at ultrahigh doping, thus preserving the Mott gap across the entire doping range from $d^1$ BNOO to $d^2$ $Ba_2CaOsO_6$ (BCOO).

The small polaron is a mobile quasiparticle composed of an excess carrier dressed by a phonon cloud[1-4]. It is manifested by local structural deformations and flat bands near the Fermi level and is significant for many applications including photovoltaics[5-7], rechargeable ion batteries[8], surface reactivity[9-11], high-$T_c$ superconductivity[12] and colossal magnetoresistance[13]. Coupling polarons with other degrees of freedom can generate new composite quasiparticles, such as magnetic[14], Jahn-Teller (JT)[15,16], ferroelectric[17] and 2D polarons[18], to name just a few. The main driving forces favoring polaron formation: phonon-active lattice, electronic correlation, and electron-phonon coupling, are realized in 3d TMOs, which represent a rich playground for polaron physics[1,19,20]. In 5d TMOs, instead, charge trapping is hindered by the large d-bandwidth and

associated weak electronic correlation, making polaron formation in a 5d orbital an unlikely event[21].

The recent discovery of SO coupled Mott insulators[22], where the gap is opened by the cooperative action of strong SO coupling and appreciable electronic correlation, has paved the way for the disclosure of novel, exciting quantum states of matter[23,24]. The coexistence of SO coupling and electronic correlation in the same TMO raises the possibility of conceptualizing a SO polaron[25,26], whose properties are determined by the complex physical scenario arising from strong SO and electron-phonon interactions. Both effects can favor or hinder polaron formation, depending on their relative strength, as shown in recent model Hamiltonian studies[27-29]. In real materials, SO polarons can be observed if a correlated relativistic

[1]Faculty of Physics and Center for Computational Materials Science, University of Vienna, 1090 Vienna, Austria. [2]Department of Physics and Astronomy, Università di Bologna, 40127 Bologna, Italy. [3]CPHT, CNRS, École polytechnique, Institut Polytechnique de Paris, 91120 Palaiseau, France. [4]Collège de France, Université PSL, 11 place Marcelin Berthelot, 75005 Paris, France. [5]Department of Mathematical, Physical and Computer Sciences, University of Parma, 43124 Parma, Italy. [6]Department of Physics, Brown University, Providence, RI 02912, USA. [7]Department of Chemistry and Biochemistry, The Ohio State University, Columbus, OH 43210, USA. ✉e-mail: cesare.franchini@univie.ac.at

background develops in a structurally flexible lattice, as in the case of the double perovskite $Ba_2NaOsO_6$ (BNOO)[30]. With a SO coupling strength $\lambda$ of 0.3 eV, a large on-site Hubbard $U$ of 3.4 eV and sizable JT vibration modes[31–34], BNOO represents the ideal candidate for questing 5d *spin-orbital polarons*.

BNOO is a Mott insulator with a low temperature canted anti-ferromagnetic (cAFM) ordered phase below 7 K, where SO splits the effective $l = 1 t_{2g}$ levels on the $Os^{7+}$ $d^1$ ion into a lower $J_{eff} = 3/2$ ground state and a doublet $J_{eff} = 1/2$[30,34,35] (see Fig. 1a). Injecting electrons in $Ba_2Na_{1-x}Ca_xOsO_6$ by chemical substitution of monovalent Na with divalent Ca ions does not cause the collapse of the Mott gap, which remains open up to full doping, when all $d^1$ sites are converted in $d^2$[36]. This indicates that excess charge carriers do not spread uniformly in the crystal, forming a metallic state, but rather should follow a different fate. Here we provide evidence that the addition of excess electrons at a local $J_{eff} = 3/2$ site produces the formation of SO/JT entangled $J_{eff} = 2$ bipolarons, which block the onset of a metallic phase.

## Results

### Polaron formation and dynamics

To gain insights on the effect of electron doping in BNOO we have performed Density Functional Theory (DFT) calculations on a $Ba_2Na_{1-x}Ca_xOsO_6$ supercell containing eight Os atoms at $x = 0.125$, corresponding to one extra electron per supercell. Figure 1a shows that the extra charge is trapped at a $d^1$ Os site, leading to a local modification of the electronic configuration from $d^1 \rightarrow d^2$. The surrounding $OsO_6$ oxygen octahedron expands isotropically by a few $10^{-2}$

Å, and new nearly flat bands develop in the mid gap region, all hall-marks of small polaron formation. This is confirmed by the local polaronic charge displayed in Fig. 1a. The delocalized alternative metal phase, with the excess charge equally distributed over all Os sites[37], is less stable than the small polaron phase by 134 meV.

Electron trapping in TMOs generally occurs in an empty $d^0$ manifold at the bottom of the conduction band, causing a $d^0 \rightarrow d^1$ transition at the trapping site associated with one mid-gap flat band. In BNOO, where the 5d orbital is singly occupied and strongly hybridized with Oxygen p states[36], we observe a conceptually different mechanism. In the undoped phase the fully occupied $d^1$ states are grouped among the topmost valence bands (see Fig. 1b) and each site contributes equally to the density of state (DOS) (green line, bandwidth $W \approx 0.3$ eV). The chemically injected excess electron goes to occupy an empty d band at the bottom of the conduction manifold, which is shifted into the gap and couples with the original $d^1$ band at the same lattice site, forming a local $d^2$ configuration (PB1 and PB2 bands in Fig. 1b) well separated by the remaining $d^1$ bands. The resulting $d^2$ dual-polaron complex can be assimilated to a bipolaron[38], as evident from the charge isosurface shown in Fig. 1a, where the PB1 and PB2 orbitals are interwoven together.

Polaron formation is confirmed by $^{23}$Na nuclear magnetic resonance (NMR) and muon spin rotation ($\mu$SR) measurements on a BNOO sample having 12.5%Ca concentration shown in Fig. 2a, b, respectively. NMR shows an anomalous peak at $T_{P,1} \approx 130$ K in the spin-lattice relaxation rate $1/T_1$ (squares), well above the temperature associated to the magnetic transition (6.8 K); correspondingly, a peak is observed in

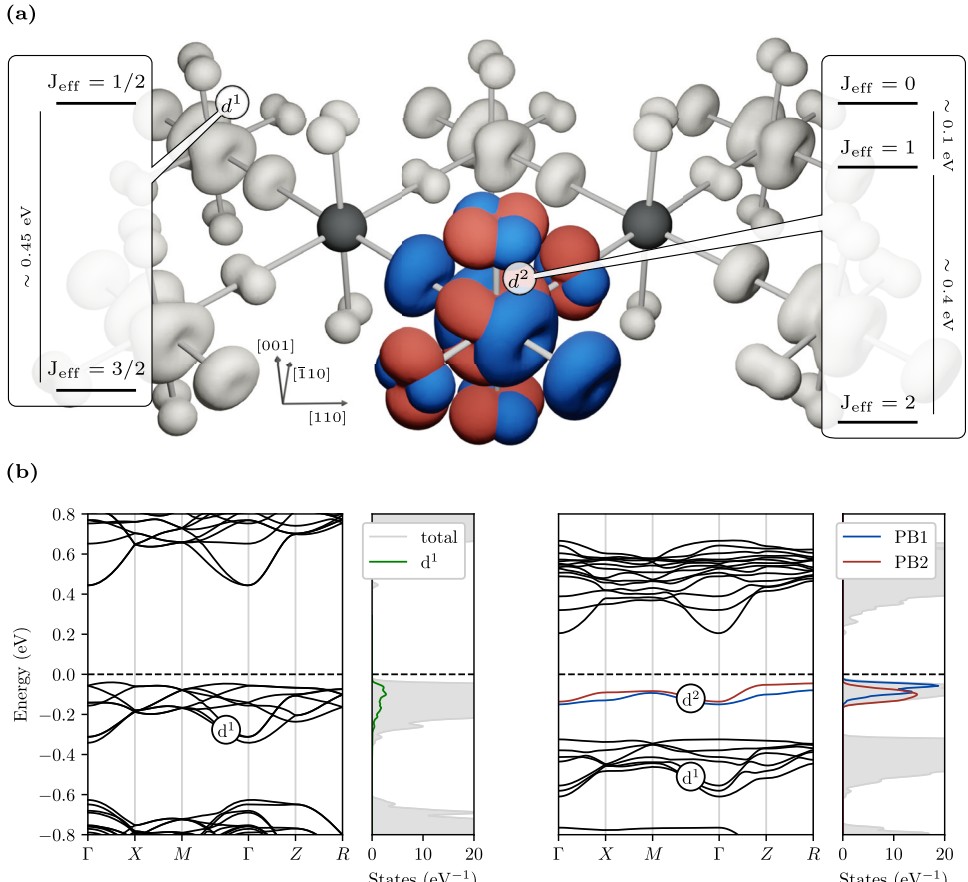

**Fig. 1 | Spin-orbit $d^2$ bipolaron in $Ba_2Na_{0.875}Ca_{0.125}OsO_6$. a** DFT charge density isosurface of the occupied Os $t_{2g}$ bands, showing the formation of $d^2$ a $J_{eff} = 2$ bipolaron coexisting with $d^1$ $J_{eff} = \frac{3}{2}$ sites characteristic of pristine $x = 0$ BNOO. Blue and red lobes refers to the entangled bipolaronic PB1 and PB2 bands displayed in (**b**). The $J_{eff}$ spin-orbital levels are obtained from DFT+HI. **b** Band structure and

relative density of states of pristine BNOO ($x = 0$, left) characterized by a multiband manifold of $d^1$ states, and bipolaronic $Ba_2Na_{0.875}Ca_{0.125}OsO_6$ (right) with a localized $d^2$ bipolaronic level below the Fermi level composed by two entangled $d^1$ bands (PB1 and PB2). The green line represents the occupied d states of a single Os site in the pristine phase.

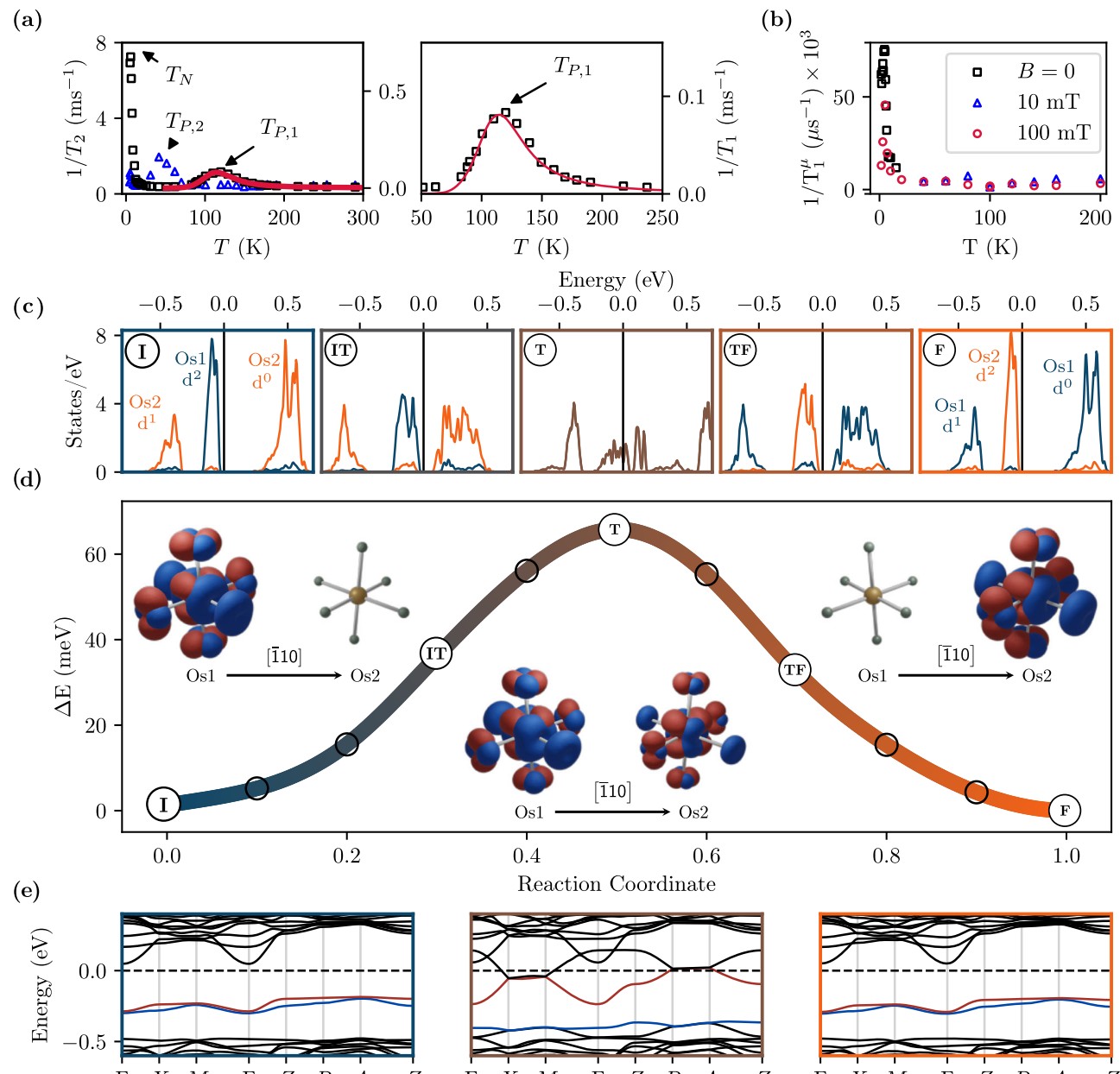

**Fig. 2 | Polaron hopping: experiment and DFT. a** NMR spin-lattice (square) and spin-spin (triangles) relaxation rates showing an anomalous peak at 130 K and 50 K due to a dynamical process. The $1/T_1$ anomalous peak is detailed in the right plot. The curve fit (red solid line) to a thermally activated BPP model provides an activation energy $E_a = 74(2)$ meV. **b** $\mu$SR data showing only a peak due to the magnetic transition but no high temperature anomalous feature corresponding to the NMR one. **c** Evolution of the density of states around the Fermi level for selected snapshots across the hopping path displayed in (**d**), projected onto the initial (I, dark blue) and final (F, orange) Os sites. The five plots correspond to reaction coordinate equal to 0.0 (I), 0.3 (IT), 0.5 (T), 0.7 (TF) and 1.0 (F). The $d^2$ bipolaron charge is

gradually transferred from the initial and final hosting sites. At the transition state (T, at 0.5) the charge is equally distributed between both Os sites giving rise to an adiabatic weakly metallic transient state (brown). **d** Potential energy for a bipolaron migrating from I to F with the charge density projected on the two neighboring Os atoms, using a color gradient from blue (bipolaron fully localized in I) to orange (bipolaron fully localized in F). The insets show the charge density isosurface decomposed over the bipolaron bands PB1 (blue) and PB2 (red). The resulting hopping barrier, 66 meV, is in excellent agreement with the experimentally-derived activation energy. **e** band structure around the Fermi level at the initial (I), transition (T) and final (F) point of the hopping process.

the spin-spin relaxation rate $1/T_2$ at $T_{P,2} \approx 50$ K (triangles). Since the fast paramagnetic fluctuations are beyond the frequency window employed and no specific magnetic interaction is expected in the explored regime[39], we attribute the NMR anomalous peaks to a charge-related thermally activated process, such as that associated with the small-polaron dynamics. This dynamical process drives electric field gradient (EFG) fluctuations which are probed by the quadrupolar interaction with the $^{23}$Na nuclear quadrupole. A peak is expected in $1/T_1$ when the frequency of the EFG fluctuations $\nu = 1/\tau_c$ (being $\tau_c$ the

fluctuation correlation time) matches the Larmor frequency (here $\omega_0 \approx 5 \cdot 10^8$ s$^{-1}$), while a peak in $1/T_2$ is anticipated when $\tau_c$ is of the order of the experimental NMR echo delay time (here of the order of microseconds). In order to confirm that the origin of the observed peaks in the NMR rates can solely be associated with the small-polaron dynamics, we have performed the $\mu$SR measurements, which are only sensitive to magnetic fluctuations. The $\mu$SR results exhibit strong critical relaxation rates at the magnetic transition temperature, 7 K, but no further relaxation peak above, in agreement with the quadrupolar

polaronic mechanism[16], since the spin 1/2 muon is not coupled to EFGs. The anomalous peak in NMR $1/T_1$ temperature dependence is fitted using a Bloembergen-Purcell-Pound-like (BPP) model for quadrupolar spin-lattice relaxation[40,41]

$$\frac{1}{T_1} = \Delta^2 \left[ \frac{\tau_c}{1+(\omega_0 \tau_c)^2} + \frac{4\tau_c}{1+(2\omega_0 \tau_c)^2} \right], \qquad (1)$$

where $\Delta^2$ is the second moment of the perturbing quadrupole-phonon coupling and the correlation time $\tau_c = \tau_0 \exp(E_a/kT)$ is expressed in terms of the activation energy $E_a$ and the characteristic correlation time $\tau_0$ of the dynamical process. The resulting fitting curve is represented by the solid line in Fig. 2a and predicts a dynamical process with activation energy of $E_a = 74(2)$ meV, $\tau_0 = 0.7(0.2)$ ps and $\Delta^2 = 2.75(6) \times 10^{10}$ s$^{-2}$.

The energy barrier extracted from $1/T_1$ is in good agreement with the activation energy predicted by DFT for a thermally activated adiabatic hopping, 66 meV, estimated at the same doping level in the framework of the Marcus-Emin-Holstein-Austin-Mott (MEHAM) theory[42,43]. The energy path between two energetically equivalent initial (I) and final (F) polaron sites Os1 and Os2 along the [$\bar{1}$10] direction, constructed with a linear interpolation scheme (LIS)[44], is shown in Fig. 2d. The hopping is a complex mechanism involving a three electron process: at the initial stage (I) the DOS (see Fig. 2c) is characterized by the d$^2$ polaron peak at the Os1 site (blue) and the unperturbed d$^1$ and d$^0$ bands at the final Os2 site (orange lines); When the hopping process starts, the Os1-d$^2$ and Os2-d$^0$ bands get progressively closer, and a fraction of the polaronic charge in Os1 transfers to the empty band in Os2. At the transition (T) state the polaron charge is equally distributed between Os1 and Os2 resulting in a local weakly metallic state (see Fig. 2c, e), as expected from an adiabatic hopping process[44]. At this point a reverse mechanism begins: the original Os2-d$^1$ and (now filled) Os2-d$^0$ merges to form a d$^2$ polaron in Os2, whereas the original Os1-d$^2$ is depleted by one electron and generates a d$^1$ band below the polaron peak. As a result, the DOS at the final point F is symmetrical to I (compare panels I and F in Fig. 2c and corresponding polaron charge isosurfaces in the insets of Fig. 2d).

## Spin-orbital polaron structure and coupling with Jahn-Teller modes

Next we investigate the nature of the polaron, unraveling an intermingled action of SO and JT-distortions in determining the energy levels and degree of stability of the polaron[45]. As single-particle approach, DFT is bound to predict *jj*-coupled levels[46], where the total angular momentum J is the vector addition of the single-electron angular momentum *j*. Indeed, the d$^2$ polaron occupation matrix computed by projecting the Kohn-Sham energy levels onto the d$^2$ polaronic subspace using spinorial projected localised orbitals[47] shows that the trapped electrons occupy two single-particle J$_{eff}$ = 3/2 levels (see Supplementary Tab. 1 and Supplementary Fig. 1) corresponding to the PB1 and PB2 bands reported in Fig. 1b. To compute the d$^2$ two-electron levels we employed Dynamical Mean-Field Theory (DMFT) within the Hubbard-I (HI) approximation applied to the DFT lattice structure relaxed with the polaronic site. This many-body approach finds that the two electrons forming the d$^2$ polaron occupy J$_{eff}$ = 2 LS-coupled levels (see Supplementary Note 2), separated from the excited J$_{eff}$ = 1 triplet by a SO gap of about 0.4 eV, as schematized in Fig. 1a. The non-polaronic d$^1$ sites (gray isosurfaces in Fig. 1a) preserves a J$_{eff}$ = 3/2 ground state, as in the pristine material[34]. Regardless the specific type of coupling, *jj* or LS, both DFT and DMFT predict spin-orbital J$_{eff}$ states, clearly indicating that the polaron is integrated into the SO-Mott background, it exhibits an individual spin-orbital state, and does not break the preexisting J$_{eff}$ = 3/2 state at the other Os sites. This leads to the coexistence of two distinct SO-Mott quantum states in the same material, a hitherto unreported physical scenario.

Although the d$^2$ polaron possesses an intrinsic spin-orbital nature, SO coupling does not play in favor of polaron formation, as inferred from the progressive increase of the polaron energy $E_{pol}$ as a function of the SO strength shown in Fig. 3a: the inclusion of SO destabilizes the polaron by about 80 meV. This behavior is linked to the effect of SO on the JT distortions recently elaborated by Streltsov and Khomskii, which suggests that for a d$^2$ configuration, SO suppresses JT distortions[45,48]. To shed light on this complex cross-coupling we have studied JT and polarons properties as a function of the effective SO coupling strength $\tilde{\lambda} = c_\lambda \lambda$ from $c_\lambda = 0.1$ to $c_\lambda = 1$ (full SO). The resulting data are collected in Fig. 3 and explained in the following.

Pristine d$^1$ BNOO exhibits a cooperative JT ordering involving the $E_g$ modes $Q_2$ and $Q_3$[34] as well as the trigonal $Q_{xy}$ mode (see Supplementary Table 4); these modes are graphically displayed in Fig. 3d–f and defined in Supplementary Table 3. Upon charge trapping, the electrostatic potential of the OsO$_6$ octahedron increases due to the additional excess charge. To counterbalance this energy cost, the oxygen cage expands according to the isotropic ($A_{1g}$) breathing-out mode $Q_1$. This expansion favors charge localization, and contributes 67% of the total $E_{pol}$ (Supplementary Fig. 5), making $Q_1$ the major lattice contribution to polaron stability. However, $Q_1$ is an isotropic deformation that does not break any local symmetry and therefore it is not related to the JT effect. Moreover, $Q_1$ is insensitive to SO and cannot play any role in the strong decrease of $E_{pol}$ with increasing $c_\lambda$ (see Supplementary Fig. 6). According to our data, only the tetragonal elongation along the [001] axis $Q_3$ is strongly dependent on $c_\lambda$ (see Supplementary Fig. 6). In particular, Fig. 3b shows that increasing $c_\lambda$ yields a continuous decrease of $Q_3$, in agreement with the analysis of ref. 45. Moreover, this SO-induced suppression of JT distortions is reflected on the JT energy ($E_{JT}$), estimated from the potential energy surface at different values of $c_\lambda$ and displayed in Fig. 3c. Therefore, $Q_3$ appears to be the key JT mode explaining the coupling between SO and polaron stability, correlating the SO-induced decrease of $E_{pol}$ with the progressive quenching of $Q_3$ and associated reduction of $E_{JT}$.

This analysis provides a new conceptual framework to interpret the complex concerted interaction between JT, SO and polaron stability. Without SO the isotropic expansion $Q_1$ and the JT modes ($Q_2$, $Q_3$ and $Q_{xy}$) help polaron stabilization providing an energy gain $E_{JT}$ which depends on the distortion amplitude. SO dampens the JT distortion $Q_3$ leading to a reduction of $E_{JT}$ and consequentially a progressive increase of $E_{pol}$ (less stable polaron) with increasing SO coupling strength. This entangled *spin-orbital Jahn-Teller bipolaron* develops in a relativistic background, is described by a spin-orbital J$_{eff}$ = 2 state and its stability is weakened by the SO-induced reduction of JT effects.

## Polaron-mediated robustness of the Mott gap at high doping

Finally, we generalize our analysis to all doping concentrations disclosing the critical role of bipolarons in preserving the Mott state and elucidating the doping-induced modulation of the polaron phonon field as measured by NMR. It is well established that a critical amount of carrier doping drives a metal-insulator transition (MIT)[49]. Formation of small polarons can delay the MIT, but at a critical polaron density, coalescence into a Fermi liquid prevails, leading to a metallic (or superconducting) phase[50–52]. Notably, electron doped BNOO represents an exception: the insulating gap remains open at any concentration as shown in Fig. 4a (the corresponding DOS are collected in Supplementary Fig. 7). This unique behavior is explained by the absence of coherent hybridization between the bipolarons, facilitated by the large Os-Os distance of ≈ 5.9 Å in the double perovskite lattice. As illustrated in Fig. 4(b), NMR shows the bipolaronic peak at $T_{P,1} \approx 130$ K at any Ca concentration with a virtually unchanged activation energy (see Supplementary Table 5), confirming the DFT results which indicate a linear increase of number of d$^2$ bipolarons with increasing doping (see Fig. 4a).

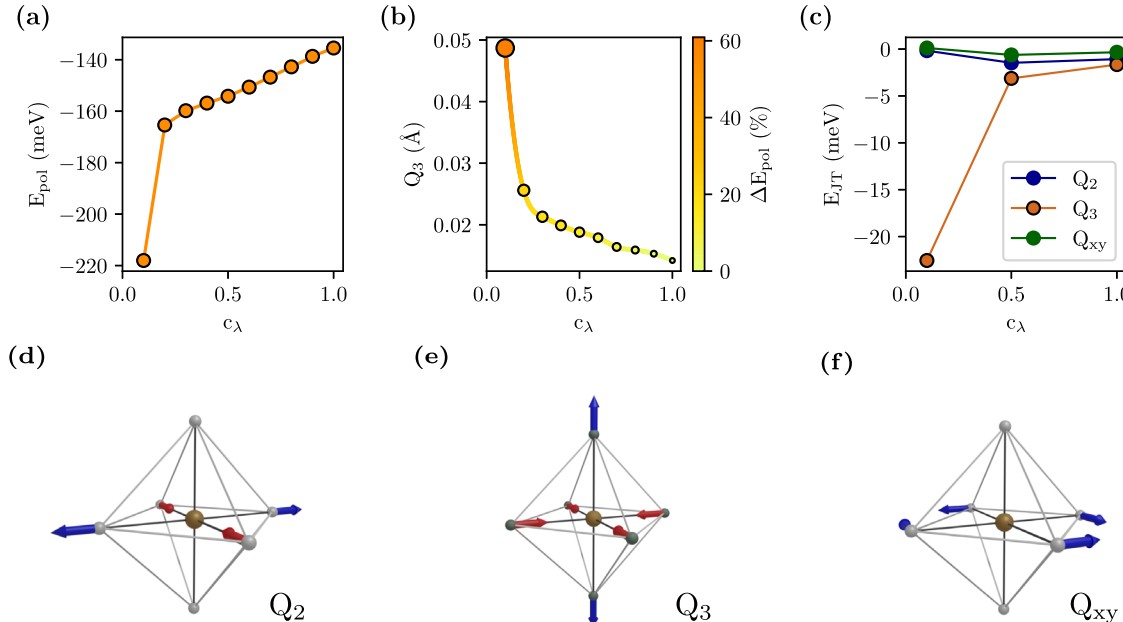

**Fig. 3 | Role of Jahn-Teller and SOC on polaron stability. a** Polaron energy $E_{pol}$ as a function of the SO coupling scaling factor $c_\lambda$ (For $c_\lambda = 0$ SO is completely suppressed, whereas $c_\lambda = 1$ refers to the full SO regime). $E_{pol}$ is progressively less negative for increasing SO coupling. **b** JT tetragonal distortion amplitude $Q_3$ as a function of $c_\lambda$. The color scale and circle's size indicate the variation of $E_{pol}$ relative to the $c_\lambda = 1$ case: SO coupling rapidly quenches $Q_3$ and reduces the polaron stability. **c** JT energy $E_{JT}$ at the polaron trapping site as a function of $c_\lambda$ for all three JT modes. $Q_3$ is the only mode influenced by SO coupling. **d**–**f** Geometrical interpretation of the JT non-zero modes $Q_2$, $Q_3$ and $Q_{xy}$ respectively.

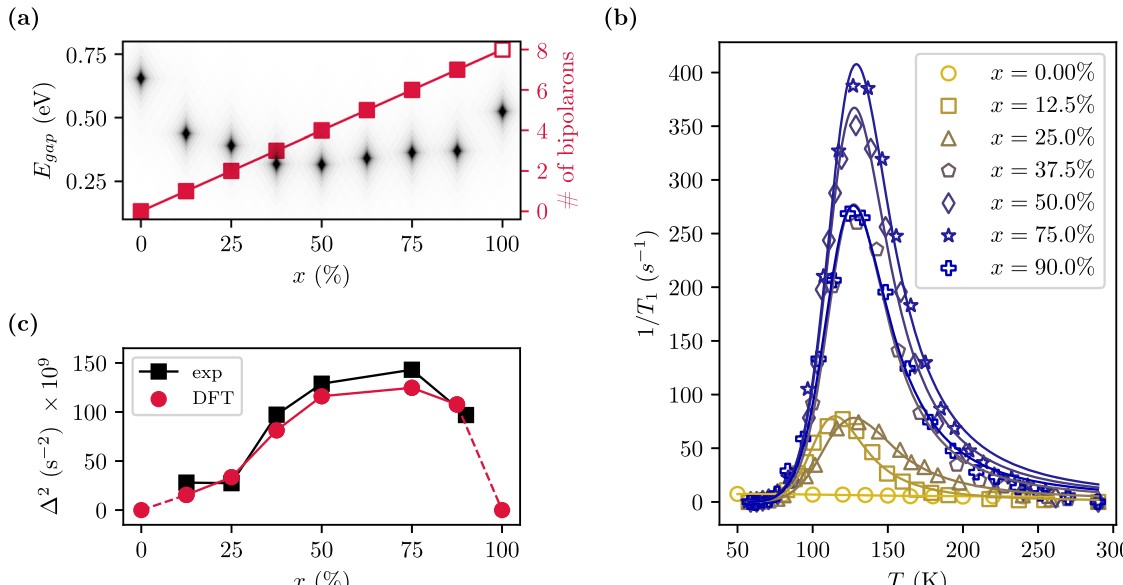

**Fig. 4 | Polaron dynamics in Ba₂Na₁₋ₓCaₓOsO₆: DFT+NMR. a** DFT energy gap (black diamonds) and number of bipolarons (red filled squares) as a function of doping. Chemically doped BNOO remains insulating for any doping concentration. The number of $d^2$ bipolaron sites grows linearly with doping. At full doping ($x = 1$, corresponding to BCOO) all Os sites are doubly occupied ($d^2$) and polaron formation is completely quenched (empty square). **b** Polaronic $1/T_1$ anomalous peak data and fitting curves calculated with the quadrupole relaxation model of Eq. (1).

For all investigated doping concentration doped BNOO exhibits a polaronic peak at approximately the same temperature $T$. **c** Second moment $\Delta^2$ of the fluctuating field as extracted from the experimental data (black squares) compared with our predicted data obtained from the DFT polaron phonon field (red circles). The dashed lines connect to pristine BNOO ($x = 0$) and BCOO ($x = 1$) where the absence of polaron leads to $\Delta^2 = 0$.

The second moment of the fluctuating field $\Delta^2$ (see Eq. (1)) as a function of doping (Fig. 4c) exhibits a dome shape, characterized by a progressive increase until $\approx 75\%$ Ca concentration, followed by a rapid decrease towards the full doping limit ($x = 1$, BCOO) with all Os sites converted into a non-polaron $d^2$ configuration in a undistorted and JT-quenched cubic lattice. To interpret these NMR measurements we have derived a model linking the polaron-induced spin-lattice relaxation rate $1/T_1$ with the oscillation of the polaron modes $\Delta Q_\xi = \langle Q_\xi \rangle_{d^2} - \langle Q_\xi \rangle_{d^1}$ (with $\xi$ running over the dominant modes $Q_1$ and $Q_3$). In particular, the average is taken over the distortions at the $d^2$ and $d^1$ sites obtained from DFT calculations. The resulting compact formula reads (see the Methods

section for a full derivation)

$$\Delta^2(x) = \frac{54(eQq_{ox})^2}{5\hbar^2 R_0^8(x)} \left[ -\Delta Q_1^2(x) + 2\Delta Q_3^2(x) \right] \quad (2)$$

where $Q$ is the quadrupole moment of the $^{23}$Na nucleus, $q_{ox}$ is the charge of the oxygen ions (as obtained by DFT, $1.78e$), and $R_0$ is the average Na-O bond length. The obtained numerical data summarized in Fig. 4c reproduce the experimental trend and indicate that, upon doping in $Ba_2Na_{1-x}Ca_xOsO_6$, the main phonon contributions to polaron dynamics are encoded in the modulation of the breathing-out mode $Q_1$ and the tetragonal distortion $Q_3$ as a function of doping. In this regard, Fig. 4c provides a transparent unprecedented microscopic interpretation of the polaron-driven spin-lattice relaxation rate $1/T_1$, here demonstrated for the quadrupolar polaron mechanism.

## Discussions

Summarizing, our study discloses a new type of polaron quasiparticle which is responsible for blocking the MIT even at ultrahigh doping and enables the coexistence of different spin-orbital $J_{eff}$ states in the same compound. This mixed-state can be interpreted as a precursor state towards the formation of the homogeneous $J_{eff} = 2$ state at full Na → Ca substitution (BCOO[53]), and the polaron is the main driving force of this transition. In perspective, this work provides the conceptual means to explore polaron physics in quantum materials with strong spin-orbit coupling including topological[54], Rashba[55] and 2D materials[56], and pave the way for polaron spintronics[57], polaron heavy-elements catalysis[58] and polaron multipolar magnetism[34,59–61].

## Methods

### Density functional theory

The electronic structure, structural deformations, and polaron hopping were studied using the fully relativistic version of VASP, employing the Perdew-Burke-Ernzerhof approximation for the exchange-correlation functional[62,63]. All DFT calculations were performed with the magnetic moments' directions fixed to those of the low-temperature cAFM phase of BNOO[34]. In addition, Dudarev's correction of DFT+U was applied to account for strong electronic correlation effects, using a value of $U = 3.4$ eV, which stabilizes the cAFM ordering in the pristine material. The computational unit cell is a $\sqrt{2}a \times \sqrt{2}a \times a$ supercell containing eight formula units, with $a = 8.27$ Å referring to the lattice constant of the standard double perovskite unit cell, which contains four formula units. The sampling of the reciprocal space was done with a k-mesh of $4 \times 4 \times 6$, and an energy cutoff of 580 eV was selected for the plane wave expansion. The SO contribution to the DFT energy functional could be manually controlled through a scaling parameter $c_\lambda$ using an in-house modified version of VASP. The analysis of the JT effect was conducted using vibration modes defined by Bersuker[64], with some minor modifications, such as neglecting the rigid translation of the octahedra and including rigid rotations (see Supplementary Note 3). Electron doping was achieved by manually increasing the number of electrons in pristine BNOO. Charge neutrality is restored by adding a homogeneous background. To extract polaronic energy levels and wavefunctions, a Wannier-like projection of Kohn-Sham wavefunctions was employed, using VASP non-collinear projected localized orbitals calculated on the polaronic Os site with the TRIQS's converter library[47] (see Supplementary Note 1). DFT calculations for comparison with NMR data were performed on relaxed chemically doped $\sqrt{2}a \times \sqrt{2}a \times a$ supercells. The same number of k-points, energy cutoff and U were used as for the previous doping method.

### Dynamical mean field theory

For the analysis of the spin-orbital structure of the polaron levels we performed charge-self-consistent DFT+DMFT calculations within the Hubbard-I approximation[65,66], using WIEN-2K[67] and the TRIQS library[68,69]. By using a $\sqrt{2}a \times \sqrt{2}a \times a$ supercell where one Na atom was substituted by one Ca (12.5% Ca concentration), we first calculated the polaronic ground state lattice structure, as explained in the previous section Density Functional Theory, using VASP. In DFT+HI calculations, the Wannier functions representing Os 5d states are constructed from the Kohn-Sham bands within the energy range $[-1, 5]$ eV around the Kohn-Sham Fermi energy, which contains the Os $t_{2g}$ and most of the $e_g$ levels. The fully-localized-limit double counting term on the polaronic Os site is set for the nominal $d^2$ occupancy, as is appropriate for the quasi-atomic Hubbard-I approximation[70], whereas for the rest of Os sites it is calculated for nominal $d^1$. The on-site interaction vertex for the full $5d$ shell is specified by the parameters $U = 3.5$ eV and $J_H = 0.5$ eV, in agreement with the previous studies of pristine BNOO[34].

### Sample preparation

Powder samples were prepared by the solid state method; stoichiometric amounts of BaO (Sigma-Alrich, 99.99% trace metals basis), CaO (Sigma-Alrich, 99.9% trace metals basis), Na2O2 (Alfa Aesar, 95%), and Os powder (Sigma-Alrich, 99.9% trace metals basis) were ground in a mortar and pestle, transferred to an alumina tube, and sealed in a quartz tube under vacuum. A separate alumina cap containing PbO2 was also included in the sealed quartz tube as the decomposition of PbO2 into PbO and O2 at 600 °C provided the oxygen source to oxidize Os metal. Because highly toxic OsO4 can form from the reaction of Os metal and O2 at or above 400 °C, this reaction was carried out inside an evacuated silica tube and the furnace was positioned in a fume hood in case the silica tube ruptured. To ensure the full oxidation of the osmium, the amount of PbO2 was chosen to generate an excess of 1/4 mol of oxygen for every mol of the desired product. The reaction vessel was heated at 1000 °C for 24 h. For several samples an additional step of grinding and heating for an additional 12 h at 1000 °C was necessary to form a homogeneous perovskite phase. See also ref. 36.

### Nuclear magnetic resonance and muon spin rotation

We exploited the nuclear spin $I = 3/2$ of $^{23}$Na nuclei in order to perform NMR spectroscopy on powder samples of $Ba_2Na_{1-x}Ca_xOsO_6$, with $x = 0.0\%$, 12.5%, 25.0%, 37.5%, 50.0%, 75.0% and 90.0%. In particular, $^{23}$Na nuclei have a sizeable quadrupolar moment that allows to probe both magnetic and charge related dynamics. We report spin-lattice ($1/T_1$) and spin-spin ($1/T_2$) relaxation rates as a function of temperature measured using an applied field of 7 T (details are reported in Supplementary Note 6). We further analysed the anomalous peaks observed in these data by implanting a beam of polarised muons spin antiparallel to their momentum into the sample and applying a magnetic field of 10 mT and 100 mT parallel to the initial muon spin polarisation in order to measure the longitudinal muon relaxation rate $\lambda_\mu \equiv 1/T_1^\mu$ (see Supplementary Note 7).

### Spin-lattice relaxation model

The interaction of the nuclear quadrupole moment with an EFG $V_{\alpha\beta}$ can be written using spherical tensor operators $T_2^q$ as[71]

$$H = \frac{eQ}{2I(2I-1)} \sum_{q=-2}^{2} (-1)^q V_q T_2^{-q} \quad (3)$$

where $Q$ is the quadrupole moment of the nucleus and $I = 3/2$ is the nuclear spin. To calculate the spherical component $V_q$ of the EFG we adopted a point-charge model of the NaO$_6$ octahedron. The explicit expressions of the $V_q$'s are given in Supplementary Note 8.

The perturbation induced by fluctuations of the Na-O bonds resulting from polaron hopping is obtained from Eq. (3) by expanding

$V_q$ in terms of the bond variations $\Delta R_{i\alpha}(t)$

$$H'(t) = \frac{eQ}{6} \sum_{i=1}^{6} \sum_{\alpha}^{x,y,z} \sum_{qq'} \mathcal{D}^2_{q'q}(\mathcal{R}) T_2^{-q} w_q^{i\alpha} \Delta R_{i\alpha}(t) \quad (4)$$

where $w_q^{i\alpha}$ are the derivatives of the spherical components of the EFG with respect to the $\alpha$-th component of the $i$-th oxygen ion. In Eq. (4) we have introduced the Wigner D-matrix $\mathcal{D}^2_{q'q}(\mathcal{R})$ to give account for the random orientation of the EFG reference frame with respect to the external magnetic field in powder samples.

The transition rate between two Zeeman levels $m$ and $m'$ averaged over all possible directions is given by

$$W_{mm'} = \frac{(eQq_{ox})^2}{6\hbar^2} \sum_q |\langle m|T_2^{-q}|m'\rangle|^2 \sum_{ij,\alpha\beta} M_{\alpha\beta}^{(ij)} \int_{-\infty}^{\infty} dt\, \overline{\Delta R_{i\alpha}(t)\Delta R_{j\beta}(0)}\, e^{-i\omega_{mm'}t}$$

$$(5)$$

where $q_{ox}$ is the oxygen ion charge in the point-charge model, $\omega_{mm'} = |\omega_m - \omega_{m'}|$ is the energy separation between the Zeeman levels, the matrices $M_{\alpha\beta}^{(ij)}$ are defined in Supplementary Note 8 and $\overline{\Delta R_{i\alpha}(t)\Delta R_{j\beta}(0)}$ is the correlation function of the $\alpha$-th component of the $i$-th bond with the $\beta$-th of the $j$-th one. To simplify the transition rate formula, some considerations on the crystal structure of $Ba_2Na_{1-x}Ca_xOsO_6$ and small polaron dynamics are necessary.

First, we notice that $NaO_6$ and $OsO_6$ octahedra are corner sharing in $Ba_2Na_{1-x}Ca_xOsO_6$. Therefore, the fluctuations of the $i$-th oxygen in the $NaO_6$ octahedron can be defined using the distortion modes $Q_\xi$ of the $OsO_6$ octahedron sharing the $i$-th oxygen with the $NaO_6$ one (see Supplementary Fig. 4). In this way the coordinate correlation functions can be expressed as correlation functions of the distortion modes $Q_{i,\xi}$ of the $i$-th $OsO_6$ octahedron.

Moving to polaron dynamics, if the $i$-th and the $j$-th octahedra are involved in an adiabatic hopping event within the time interval $-\tau_c \lesssim t \lesssim \tau_c$, in the LIS we have $Q_{i,\xi}(t) = Q_{j,\xi}(-t)$. Thus, assuming that independent modes at the same site are uncorrelated, we can write all the correlation functions appearing in Eq. (5) as autocorrelation function of the independent distortion mode at each $OsO_6$ octahedron $\overline{Q_\xi(t)Q_\xi(0)}$. To calculate these quantities, we recall that the MEHAM theory of adiabatic small polaron hopping relies on the classical treatment of phonon modes in the description of the site-jump process[72]. In this limit, one can describe fluctuations using a phenomenological Langevin equation[73]. Within this model, if the characteristic time of the fluctuations is much smaller than that of the hopping process ($\tau_c$), we speak of overdamped regime and the fluctuations' autocorrelations are given by

$$\overline{Q_\xi(t)Q_\xi(0)} = \Delta Q_\xi^2\, e^{-|t|/\tau_c} \quad (6)$$

where $\Delta Q_\xi$ is the amplitude of the fluctuations of the $Q_\xi$ mode. To validate this assumption, we evaluated the vibration frequencies $\omega_Q$ of the $Q_1$ and $Q_3$ oscillators from the potential energy curves obtained in the analysis of the Jahn-Teller modes (reported in Supplementary Note 3). We found them to be in the order of $\omega_Q \sim 10^{13}$ s$^{-1}$, as also reported in phonon spectra calculated by Voleti et al.[60]. On the other hand, the correlation times extracted from NMR measurements are in the order of $\tau_c \sim 10^{-10}$ s, therefore $\omega_Q \tau_c \gg 1$, which corresponds to the overdamped regime[73]. Moreover, we notice that the behavior of the autocorrelations expressed in Eq. (6) is commonly assumed in the description of spin-lattice relaxation processes[40,71,74].

By neglecting correlation functions between opposite sites (Os-Os distance ≈ 8.29 Å), i.e. assuming only nearest-neighbor hopping (Os-Os

distance ≈ 5.86 Å), the transition rate becomes

$$W_{mm'} = \frac{3(eQq_{ox})^2}{2\hbar^2 R_0^8} \sum_q |\langle m|T_2^q|m'\rangle|^2 \left(-\Delta Q_1^2 + 2\Delta Q_3^2\right) \frac{\tau_c}{1+(\omega_{mm'}\tau_c)^2} \quad (7)$$

where we have only considered fluctuations in the breathing-out mode $Q_1$ and the tetragonal mode $Q_3$ to be relevant, as deduced from the considerations expressed in the main text.

The spin-lattice relaxation time $T_1$ for a $I = 3/2$ nucleus with quadrupolar interactions is given by[71]

$$\frac{1}{T_1} = \frac{12}{5}(W_1 + 4W_2) \quad (8)$$

where $W_1$ and $W_2$ are the transition rates for relaxations with selection rules $\Delta m = \pm 1$ and $\Delta m = \pm 2$ respectively. By combining the latter Eq. (8) with the transition rate formula in Eq. (7), we obtain the relation

$$\frac{1}{T_1} = \Delta^2 \left[\frac{\tau_c}{1+(\omega_0\tau_c)^2} + \frac{4\tau_c}{1+(2\omega_0\tau_c)^2}\right] \quad (9)$$

which has been used to fit the NMR anomalous peak $1/T_1(T)$ with $\tau_c = \tau_0 \exp(T_a/T)$, while $\Delta^2$ is given by

$$\Delta^2 = \frac{54(eQq_{ox})^2}{5\hbar^2 R_0^8}\left(-\Delta Q_1^2 + 2\Delta Q_3^2\right) \quad (10)$$

which corresponds to Eq. (2) of the main text. Based on the above analysis and in analogy with the standard BPP model, $\Delta^2$ corresponds to the second moment of the fluctuating perturbation $H'(t)$ which is expressed in terms of the amplitude of the distortion modes.

## Data availability

Theoretical and exeprimental data are collected in the Supplementary Notes. The DFT structural data (POSCAR files) used in this study have been deposited in the PHAIDRA database under accession code https://phaidra.univie.ac.at/o:2045864.

## Code availability

DFT first-principles calculations were performed using the licensed VASP code. The modifications to the source code necessary to control the SO coupling strength are available upon request to the authors. The licensed code WIEN2K and the free library TRIQS were used for DFT+DMFT calculations.

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

## Acknowledgements

L.C. thanks Michele Reticcioli and Luigi Ranalli for useful discussions. Support from the Austrian Science Fund (FWF) projects I4506, J4698 and SFB-F81 TACO (C.F.) is gratefully acknowledged. L.C. and D.F.M. acknowledge the Vienna Doctoral School of Physics. The computational results have been achieved using the Vienna Scientific Cluster (VSC). This work was supported in part by U.S. National Science Foundation (NSF) grant No. DMR-1905532 (V.F.M.), the NSF Graduate Research Fellowship under Grant No. 1644760 (E.G.), NSF Materials Research Science and Engineering Center (MRSEC) Grant No. DMR-2011876 (P.M.T and P.M.W.). This work is based on experiments performed at the Swiss Muon Source SuS, Paul Scherrer Institute, Villigen, Switzerland. For the purpose of open access, the author has applied a CC BY public copyright licence to any Author Accepted Manuscript version arising from this submission.

## Author contributions

C.F. conceived and supervised this project. L.C. executed all DFT calculations and analyzed the results, assisted by D.F.M. L.V.P. conducted the DMFT calculations. S.S. and V.F.M. have conceived and coordinated the experimental activity. P.M.T. and P.M.W. prepared the samples. G.A., A.T., P.C.F., R.C. and E.G. performed nuclear magnetic resonance measurements and analysis. R.D.R., R.C., E.G. performed muon spin spectroscopy measurements and analysis. L.C. developed the spin-lattice relaxation model with inputs by C.F., G.A. and R.D.R. C.F., L.C. and S.S. wrote the manuscript with input from all the authors.

## Competing interests

The authors declare no competing interests.
