## [Peer Review File · Nature Communications]

REVIEWER COMMENTS

Reviewer #1 (Remarks to the Author):

Authors have investigated the electronic structure of $\text{Ba}_2\text{Na}_{1-x}\text{Ca}_x\text{OsO}_6$ ($0 < x < 1$) by using ab initio calculations. They claim that the formation of SO/JT entangled bipolarons is the key to understanding the insulating state even at the high doping level ($x = 1$). I think this result is essential for the study of bipolarons and spin-orbit coupling in solids. However, the authors' claim might be useful only for a part of solid materials. In my opinion, the present manuscript does not seem to be so novel and does not deserve publication in Nat. Comm. at the present stage. I have some questions and comments regarding the authors' claim:

1) There are already many reports on SO coupling and small polarons. Thus, research solely on SO interaction and polarons does not offer sufficient novelty. The authors should more concretely show the specific materials for which the spin-orbit Jahn-Teller bipolarons are effective.

2) In Fig. 3, the authors discuss the relation between the effective SO coupling strength and E_{JT} for BNOO. Is this discussion valid even for $x > 0$? The symmetry of the system should, at least, be broken when Ca is doped, so this discussion seems to be invalid. In particular, the energy difference between Q2, Q3, and Qxy is only about 20 meV, so another factor seems to be more important when Ca is doped.

3) In Fig. 4, the intensity of $1/T_1$ changes largely between $x = 0.25$ and 0.375 . Why does this happen?

4) The authors claim that the crystals are insulators in the whole range of x from 0 to 1. What is the resistance of the samples ($x = 0.00 - 0.90$) used by the authors? How closely does the band gap, as measured by, e.g., optical methods, consist of the calculated value shown in Fig. 4(a)?

Minor comments:

a) To avoid reader misunderstanding, the notation of "l"- "F" should be consistent between Fig. 2(c) and (d). In Fig. 2(c), "l" corresponds to a reduction coordinate of 0.1, but in Fig. 2(d), "l" corresponds to 0.0.

b) I would like to ask the authors to show the total DOS in Fig. 7S in a similar way to that shown in Fig. 1(b).

Reviewer #2 (Remarks to the Author):

This paper reports NMR, μ SR, and DFT calculations of the $\text{Ba}_2\text{NaOsO}_6$ in which the Na is substituted with Ca. The behavior of the parent compound is driven by the Os 5d1 states, which experience a large spin orbit interaction and strong correlations that drive Mott insulating behavior. This behavior is manifest as a gap that surprisingly is not suppressed when doping with Ca to introduce electrons. The authors report Na-23 NMR data on polycrystalline samples which shows a peak in $1/T_1$ emerges near 140K upon doping, that is only slightly dependent on the doping level. On the other hand, μ SR data show no evidence for such a peak. The authors conclude that the origin of the peak arises from quadrupolar relaxation that couples to the spin 3/2 Na-23 nucleus, but not to the spin 1/2 muon. Detailed electronic structure calculations indicate that the lattice distorts in the vicinity of the Os in a 5d2 configuration, forming a mobile bipolaron. The hopping of this entity is thermally activated, giving rise to fluctuating electric field gradients that drive the Na-23 T_1 .

The manuscript is well-written and presents a reasonable scenario to explain the NMR and μ SR data. I recommend publication after the authors consider the following comments.

[1] The NMR T_1 recovery data were fit to a stretched exponential form (Eq. 8S) for a spin-1/2 nucleus. However, a major claim is that the recovery is driven by quadrupolar fluctuations, which should exhibit a different relaxation form. Moreover, the nucleus in question has spin 3/2. It is therefore somewhat questionable whether Eq (8S) is appropriate, although it likely captures the relevant physics. It would also be good to include the relaxation form for the muons. In particular, does the muon relaxation also exhibit stretched behavior?

[2] It might be worthwhile to analyze the relaxation data in terms of the inverse Laplace transform approach, e.g. as in doi: 10.1103/PhysRevB.101.174508, which uncovered a contribution from the charge dynamics.

[3] It would be worthwhile to provide a reference for, or show, any transport or optical data that indicate how the gap behaves as a function of doping, if possible.

[4] Eq. (9) demonstrates that the T1 peak for the NMR data should exhibit a field dependence. Do the authors have any data that would support this interpretation?

[5] Fig. 1a shows isosurfaces the two bipolaron bands. These bands are presumably dispersive and delocalized, but the figure shows the red and blue surfaces only localized on a single site. Is what shown actually a type of Wannier function for the bipolaron?

Reviewer #3 (Remarks to the Author):

Report on "Spin-orbital Jahn-Teller bipolarons" by Lorenzo Celiberti et. al.

This work deals with bipolaron formation in 3d transition metal oxides. It presents both a experimental analysis from NMR and μ SR and theoretical DFT calculations.

Authors find a doping-induced bipolaronic phase associated with Jahn-Teller deformation. Bipolarons has a spin-orbital character associated with a peculiar total angular momentum. This finding is supported by NMR measurement as a function of temperature showing strong coupling to the lattice at temperatures larger than the magnetic transition temperature which is insted probed by μ SR.

The DFT calculations suggest the formation of a bipolaronic non-ordered phase which prevents the system being metallic at large dopings. Further NMR study confirm the stabilisation of a bipolaronic phase upon doping.

I think that the main results presented in this paper are convincing and my overall judgment on this work is positive.

I however there are some points which must be considered by the authors expecially in view of the broad audience of this journal.

major items

- In fig. 1 (b) the d^2 band has a sizeable dispersion of about 0.1 eV. It is well known that if a strong electron-lattice interaction is able to form a polaronic or bipolaronic state this state is almost undispersed and prone to a real localisation around impurities. Could the authors comment on the bandwidth of the bipolaronic state they find and on possible action fo disorder if any in this material.

- In TMO the large local Hubbard repulsion is at the origin of the insulating character. It is well known that large local Hubbard repulsion do *compete* with *singlet* bipolaron formation and instead could allow for a *polaronic* state. The DMFT treatment adopted in this work do allow in principle the author to take into account this effect. It is therefore non-trivial to explain to a broad audience why and how a possibly non-singlet bipolaronic state can be preserved despite the large value of U .

- As authors' state in the abstract spin-orbit and and polaron formation is "seemingly mutually exclusive". For a broad audience reader this statement it is not really obvious. Explanation indeed comes later in the discussion and it is far less trivial. I suggest the authors to take into account the literature on simple models discussing the interplay between electron-lattice and spin-orbit interaction.

In simple models where the electron phonon-interaction is purely local (Holstein) a sort of competition was found between localization effects and spin-orbit by [Covaci et al.](<https://journals.aps.org/prl/abstract/10.1103/PhysRevLett.102.186403>). However this competition may turn into cooperation even for local electron-lattice interaction (see [Cappelluti et al.](<https://journals.aps.org/prb/abstract/10.1103/PhysRevB.76.085334>)) or Frölich (see [Grimaldi](<https://journals.aps.org/prb/abstract/10.1103/PhysRevB.81.075306>)).

minor changes

- Fig. 2(a) hard to read symbols are too big. I suggest to split the figure and the inset in two in order to enlarge the y scale of the main panel and improve readability.

- IMT for Metal to Insulator Transition is non standard I suggest MIT.

“*NCOMMS-23-34580-T: Spin-orbital Jahn-Teller bipolarons*”

— Reply to the Reviewers

Reply to Reviewer 1

Referee: I think this result is essential for the study of bipolarons and spin-orbit coupling in solids. However, the authors’ claim might be useful only for a part of solid materials. In my opinion, the present manuscript does not seem to be so novel and does not deserve publication in Nat. Comm. at the present stage.

Authors: We thank the referee for considering our work essential for the solid state community working on polarons. We believe our work is the first study providing clear evidence and materials-specific characterisation of the spin-orbital polaronic quasiparticle as well as its role in freezing the Hubbard gap thus blocking the expected doping-induced insulator-to-metal transition (MIT). To achieve this results we have employed a wide array of computational and experimental approaches and developed a model to interpret the NMR data with DFT-derived polaron distortions field. Therefore we believe that our study will attract the interest of a large community of scientists working on polaron physics, MIT, Hubbard(-Dirac) insulators, spin-orbit coupling effects and Jahn-Teller physics from a computational, theoretical and experimental perspective. We underline that in the last few years there have been a surge of interests in SOC-driven effects and in polaron physics, with many articles published in highly-reputed journals including Nature Communication [1–6].

Referee: There are already many reports on SO coupling and small polarons. Thus, research solely on SO interaction and polarons does not offer sufficient novelty. The authors should more concretely show the specific materials for which the spin-orbit Jahn-Teller bipolarons are effective.

Authors: We agree with the referee that there are many reports individually studying polarons or SOC effects. To the best of our knowledge, there have been no prior studies that combine ab-initio calculations and theoretical analysis with experimental methods to establish and understand the presence and role of small polarons coupled with SOC in a material. Through a many-body based analysis of the multiplet level structure we have deciphered in a transparent way the nature of the SOC polaron. Moreover, our research extends beyond SO interactions and polarons. It explores the interplay among SO coupling, JT effect, and polarons and the role of polarons in phenomena like NMR spin-lattice relaxation and the metal-insulator transition. There have been previous model Hamiltonians studies that have explored the possible interaction between polaron and SOC, which we have added and commented in the introduction: 10.1103/PhysRevLett.102.186403, 10.1103/PhysRevB.76.085334, 10.1103/PhysRevB.81.075306.

Referee: In Fig. 3, the authors discuss the relation between the effective SO coupling strength and E_{JT} for BNOO. Is this discussion valid even for $x > 0$? The symmetry of the system should, at least, be broken when Ca is doped, so this discussion seems to be invalid. In particular, the energy difference between Q2, Q3, and Qxy is only about 20 meV, so another factor seems to be more important when Ca is doped.

Authors: One of the main message of our work is that we were able to study the evolution of the SOC-polaron properties in the full doping regime from $x=0$ to $x=1$.

The referee correctly notes that Ca doping disrupts the system’s symmetry. However, this disruption does not pose a significant challenge to our analysis of the competition between SO and JT effects, which in fact relies on a JT-impurity model. We justify this model by observing that polaronic distortions typically remain within a single OsO_6 octahedron and can thus be described as an impurity center featuring JT instability, as discussed in the main text. In particular, within this framework, the modes Q_i describe local distortions within the OsO_6 octahedron.

Depending on the positions of the polaron and Ca ions, the values of Q_i may vary, particularly if the CaO_6 and polaronic OsO_6 octahedra share a corner. However, it's important to note that the SO-JT competition generally arises from differing orbital symmetries preferred by these two effects (*e.g.* see Ref. [7]). Therefore, even with Ca doping-induced changes in the value or symmetry of the relevant modes, we expect the qualitative behavior of the distortions with respect to SO to remain consistent.

The Reviewer correctly points out that the JT energy gain of 20 meV seems to be too small for being relevant. In fact, as we report in the main text, the primary lattice contribution to polaron stability comes from the Q_1 mode, representing the isotropic expansion of the octahedron to accommodate the extra charge of the polaron. In this regard, the JT energy gain of 20 meV resulting from the Q_3 distortion at low SO should be considered as a supplementary factor to this isotropic expansion. Therefore, the JT should not be considered the primary driving force for polaron stabilization but rather an additional factor that supports the formation of polarons.

Referee: In Fig. 4, the intensity of $1/T_1$ changes largely between $x = 0.25$ and 0.375 . Why does this happen?

Authors: According to Eq. 2 the intensity of $1/T_1$ is correlated to the fluctuating distortion field. To interpret the experimental data we have computed the amplitude of these fluctuations as a function of doping. Our DFT data show that at $x = 0.25$ there is a steep increase of the polaronic distortions ΔQ_1 and ΔQ_3 , which causes the large changes mentioned by the Reviewer.

Referee: The authors claim that the crystals are insulators in the whole range of x from 0 to 1. What is the resistance of the samples ($x = 0.00 - 0.90$) used by the authors? How closely does the band gap, as measured by, *e.g.*, optical methods, consist of the calculated value shown in Fig. 4(a)?

Authors: We thank the Reviewer for rising this important issue. Indeed, measurements of the resistance and band gap of the considered compounds would be an important complement to our analysis. However, to the best of our knowledge, neither optical conductivity data nor transport measurements are available in the literature for these materials. Unfortunately, reliable transport measurements would require yet unavailable single crystals, to avoid inter-grain resistance contributions. Nevertheless, we have indirect evidences on the insulating behaviour of our samples. In all our measurements, we consistently observe that the Q factor of the NMR resonating circuits remains constant across varying Ca concentrations, suggesting a high resistivity for all samples. This observation is supported by DFT calculations, which predict the insulating phase to be energetically more favorable than the metallic one. While DFT calculations cannot replace experimental evidence, we have verified the validity and robustness of our DFT approach against several measured properties. Specifically DFT reproduces well: (i) The Os L3 XANES spectra and the corresponding doping-dependent variation of the $t_{2g}-e_g$ splitting [8]; (ii) The magnetic ordering and in particular the canting angle of the ground state canted-AFM state of the pure Na compound [9]; (iii) Structural properties, in particular the symmetry broken phase (in agreement with NMR) [1, 9] as well as the doping dependent volume expansion [8]; (iv) EFG parameters of the nuclear quadrupolar coupling [10].

Therefore, we are confident on the insulating nature of all considered samples, and plan to confirm this conclusion with future measurements on single crystals (when they will become available).

Referee: To avoid reader misunderstanding, the notation of "I"-F" should be consistent between Fig. 2(c) and (d). In Fig. 2(c), "I" corresponds to a reduction coordinate of 0.1, but in Fig. 2(d), "I" corresponds to 0.0.

Authors: The notation "I"-F" corresponds always to reaction coordinates 0.0 and 1.0 respectively. We have corrected the typos in the caption of Fig. 2. Thank you for pointing this out.

Referee: I would like to ask the authors to show the total DOS in Fig. 7S in a similar way to that shown in Fig. 1(b).

Authors: We report below the DOS of Fig. 7S including the total DOS. We have correspondingly modified Fig. 7S in the Supplementary Informations.

Figure 1: DOS projected onto the d-orbitals of Os d^1 (blue) and d^2 (orange) sites at different Ca concentration x . The total DOS is shown in grey.

Reply to Reviewer 2

Referee: The manuscript is well-written and presents a reasonable scenario to explain the NMR and μ SR data. I recommend publication after the authors consider the following comments.

Authors: We thank the referee for appreciating our study and for recommending its publication in a suitably revised form.

Referee: The NMR T_1 recovery data were fit to a stretched exponential form Eq. (8S) for a spin-1/2 nucleus. However, a major claim is that the recovery is driven by quadrupolar fluctuations, which should exhibit a different relaxation form. Moreover, the nucleus in question has spin 3/2. It is therefore somewhat questionable whether Eq. (8S) is appropriate, although it likely captures the relevant physics. It would also be good to include the relaxation form for the muons. In particular, does the muon relaxation also exhibit stretched behavior?

Authors: We thank the Reviewer for raising this point and for thus giving us the opportunity to better clarify an important aspect of our work. The stretched exponential β in Eq. (8S) is indeed commonly used for spin-1/2 systems with a distribution of relaxation times ($\beta < 1$). However, the model can be extended to higher nuclear spins if a single exponential recovery behaviour is expected; this is typically the case for cubic crystal with vanishing static quadrupolar interactions and spin-spin interactions that maintain a spin temperature [11, 12]. These conditions are satisfied in the temperature range considered in our work (50-300 K in Fig. 4(b)), since a single NMR line with 10-30 kHz linewidth (as shown in Fig. 5 of Ref. [13]) is observed. Residual static quadrupolar interactions due to static local defects would produce the characteristic non monotonic recoveries predicted by Andrews and Tunstall (shown in Fig. 4 of Ref. [12]). However, we do not observe such non monotonic recoveries. Deviations from the simple exponential behaviour ($\beta = 1$) occur close to the $1/T_1$ peak, where β approaches the value 0.5 (see Fig. 10S). This typically indicates the presence of a distribution of spin lattice relaxation times, commonly observed for both extrinsically and intrinsically disordered systems [14–16]. In our case, the exponent β indicates that disorder reaches its maximum on cooling towards the crossover between dynamic and static conditions on the NMR time scale, at the $1/T_1$ peak. This typically occurs in the presence of intrinsic electronic inhomogeneity (see *e.g.* Fig. 1 in Ref. [17], referring to a compound undergoing an electronic nematic transition, and Ref. [16]), which in $\text{Ba}_2\text{Na}_{1-x}\text{Ca}_x\text{OsO}_6$ arises from the freezing process of polaronic injected charges (see also Ref. [13]).

In contrast, the muon longitudinal relaxation, reported in Fig. 11S, obeys the simple exponential form of Eq. (11S) in the whole temperature range (see also Ref. [13]), since the muon does not experience quadrupolar interactions and is insensitive to the charge inhomogeneity.

We have rephrased and added the following remarks below Eq (8S) and Fig. 10S to clarify these aspects.

- Below Eq. 8S. We have replaced:

"where the common interpretation of β is, in terms of the global relaxation, a system containing many independently relaxing species, resulting in the sum of different exponential decays.

The stretched behaviour is often observed in complex transition metal oxides. It typically reflects the presence of a non trivial distribution of relaxation rates due to local electronic inhomogeneities, which give rise to a site dependent magnetic or electric coupling.

A single exponential decay is clearly corresponding to $\beta = 1$, while $\beta = 0.5$ is typical of a fully disordered system, *i.e.* an intrinsic heterogeneity of phases, which can be described with a multi-exponential factor."

with

"where the common interpretation of $\beta < 1$ is, in terms of the global relaxation, that the sample contains many independently relaxing species, resulting in the sum of different exponential decays. This is expected in systems with considerable electronic inhomogeneity, giving rise to a broad distribution of correlation times [14, 16, 17]."

- Below Fig.10S we have replaced

"Notice that the stretching coefficient is reduced to $\beta = 0.5$ at the relaxation rate peaks, revealing an electronic inhomogeneity *i.e.* a T_1 distribution, but it approaches an exponential relaxation with $\beta = 1$ elsewhere. The latter agrees with a quadrupolar relaxation mechanism as shown in Fig. 4 and Sec. 5S."

with

"Notice that the stretching coefficient is reduced to $\beta = 0.5$ at the relaxation rate's peak, which indicates that the electronic disorder reaches its maximum when cooling towards the crossover between the dynamic and the static regime on the NMR time scale, but it approaches a single exponential decay with $\beta = 1$ elsewhere. The single relaxation rate is expected also for quadrupolar nuclei ($I > 1/2$) in a cubic crystal since the static quadrupolar interaction vanishes and spin-spin interactions maintain a spin temperature [11] [12]. Indeed, these conditions are satisfied throughout the reported temperature range, where a single NMR line is observed, with linewidth below 30 kHz [13]."

Referee: It might be worthwhile to analyze the relaxation data in terms of the inverse Laplace transform approach, e.g. as in doi: 10.1103/PhysRevB.101.174508, which uncovered a contribution from the charge dynamics.

Authors: We appreciate the reviewer’s suggestion to incorporate the Inverse Laplace Transform (ILT) technique into our data analysis. The ILT technique is known for effectively revealing details in correlation time distributions, especially in distinguishing bimodal distributions in more disordered cases. Implementing ILT requires a considerable time investment and the execution of additional long time-scale measurements for data acquisition. However, considering that our specific case of polaron diffusion does not typically exhibit multimodal components, we posit that an ILT analysis might not offer substantial additional insights. We propose using the logarithmic mean of the ILT, known to be equivalent to the stretched rate parameter [18, 19], which we believe adequately serves our current study’s objectives. Moreover, our chosen approach of combining the analysis of NMR rate distribution with muon longitudinal relaxation, insensitive to charge inhomogeneities, provides comprehensive evidence supporting the conclusions in our manuscript.

In summary, while we value the recommendation to use ILT, practical considerations and the unique characteristics of our system lead us to favor the use of the logarithmic mean of the ILT alongside our established analytical framework. This approach maintains the robustness of our scientific findings with less emphasis on the ILT technique.

Referee: It would be worthwhile to provide a reference for, or show, any transport or optical data that indicate how the gap behaves as a function of doping, if possible.

Authors: To the best of our knowledge, there is no transport and/or optical data on our powder samples. This is for the reasons explained in our answer to Referee 1. We report below the response given to Reviewer 1:

We thank the Reviewer for rising this important issue. Indeed, measurements of the resistance and band gap of the considered compounds would be an important complement to our analysis. However, to the best of our knowledge, neither optical conductivity data nor transport measurements are available in the literature for these materials. Unfortunately, reliable transport measurements would require yet unavailable single crystals, to avoid inter-grain resistance contributions. Conductivity measurements in bulk polycrystals often pose challenges in discerning the intrinsic band gap, the polaron hopping barrier, and the spurious intergrain contribution, as illustrated in Figure 3 of Ref. /citeD3DT00381G, for example. Nevertheless, we have indirect evidences on the insulating behaviour of our samples. In all our measurements, we consistently observe that the Q factor of the NMR resonating circuits remains constant across varying Ca concentrations, suggesting a high resistivity for all samples. This observation is supported by DFT calculations, which predict the insulating phase to be energetically more favorable than the metallic one. While DFT calculations cannot replace experimental evidence, we have verified the validity and robustness of our DFT approach against several measured properties. Specifically DFT reproduces well: (i) The Os L3 XANES spectra and the corresponding doping-dependent variation of the $t_{2g}-e_g$ splitting [8]; (ii) The magnetic ordering and in particular the canting angle of the ground state canted-AFM state of the pure Na compound [9]; (iii) Structural properties, in particular the symmetry broken phase (in agreement with NMR) [1, 9] as well as the doping dependent volume expansion [8]; (iv) EFG parameters of the nuclear quadrupolar coupling [10].

Therefore, we are confident on the insulating nature of all considered samples, and plan to confirm this conclusion with future measurements on single crystals (when they will become available).

Referee: Eq. (9) demonstrates that the T1 peak for the NMR data should exhibit a field dependence. Do the authors have any data that would support this interpretation?

Authors: We thank the Reviewer for suggesting to check predicted field dependence from Eq. (9) of the manuscript. We have tested the predicted field dependence by comparing the temperature dependence of the NMR rates ($1/T_1$) measured at two different applied magnetic

fields, *i.e.* at 1.4 T and 7 T for the sample $\text{Ba}_2\text{Na}_{1-x}\text{Ca}_x\text{OsO}_6$ with $x = 0.5$. These results and respective fits to the BPP temperature dependence (represented by the solid lines) are plotted in the figure below. We point out that because the NMR signal should increase as square of the applied field, signal-to-noise ratio for the data at 1.4 T decreases significantly, leading to appreciable larger error bars in $1/T_1(T)$ data measured at 1.4 T.

Figure 2

As evident from the plot above, the two fit curves scale well with the predicted field dependence from Eq. (9). Specifically, by performing the fit to Eq. (9) for applied fields of 1.4 T and 7 T, we found the following fitting parameter values: $E_a = 900$ K, $\Delta^2 = 130 \cdot 10^9 \text{ s}^{-2}$ and $\tau_0 = 1.5$ ps with small variation within ± 50 K, $70 \cdot 10^9 \text{ s}^{-2}$ and 0.5 ps, respectively.

Referee: Fig. 1a shows isosurfaces the two bipolaron bands. These bands are presumably dispersive and delocalized, but the figure shows the red and blue surfaces only localized on a single site. Is what shown actually a type of Wannier function for the bipolaron?

Authors: One of the primary characteristics of polaron formation is the presence of localized bands with minimal dispersion, as observed in various studies (e.g., see Ref. [20]). In Fig. 1a, the isosurfaces are derived from the charge density decomposition of the bands. Specifically, we select the polaronic Kohn-Sham bands PB1 and PB2, following the methodology outlined in the VASP manual available at this link and plotting the associated charge density at the constant value of 0.001 \AA^{-2} . This analysis does not involve the use of Wannier functions, which, however, would deliver the same outcome.

Reply to Reviewer 3

Referee: I think that the main results presented in this paper are convincing and my overall judgment on this work is positive.

Authors: We thank the referee for the positive judgment of our work and for considering our results interesting and convincing.

Referee: In fig. 1 (b) the d^2 band has a sizeable dispersion of about 0.1 eV. It is well known that if a strong electron-lattice interaction is able to form a polaronic or bipolaronic state this state is almost undispersed and prone to a real localisation around impurities. Could the authors comment on the bandwidth of the bipolaronic state they find and on possible action

for disorder if any in this material.

Authors: The Reviewer is correct in pointing out that polaron formation results in an almost undispersed state. However, in real materials the localization is never perfect due to hybridization with neighbouring sites (in particular neighbouring oxygen ions) and therefore the polaronic states present a small dispersion. Various DFT studies on small polarons and bipolarons have shown this effect in different oxides, *e.g.*: BaBiO₃ [20], KTaO₃ [6], WO₃ [21] and BiVO₄ [22]. We did not consider disordered structures in our calculations.

Referee: In TMO the large local Hubbard repulsion is at the origin of the insulating character. It is well known that large local Hubbard repulsion do *compete* with *singlet* bipolaron formation and instead could allow for a *polaronic* state. The DMFT treatment adopted in this work do allow in principle the author to take into account this effect. It is therefore non-trivial to explain to a broad audience why and how a possibly non-singlet bipolaronic state can be preserved despite the large value of U .

Authors: The Reviewer raises a very important point regarding the competition between Coulomb repulsion and electron-phonon coupling in bipolarons formation. This problem has been first formulated by Emin in Ref. [23] within the molecular crystal model, where one considers the problem of two electrons on a lattice, that can either localize in "single polaron" states at different sites or coalesce in a small bipolaron on a single site if the structural energy gain is larger than the repulsive Coulomb interactions, as the referee has correctly pointed out. However, the situation encountered in our study is different as the excess injected electron couple with a pre-existing 5d¹ state forming a bipolaronic state. Within this scenario there is no way to form two separated single polaron states: each excess electron is trapped in a 5d¹ forming a bipolaron. This situation is qualitatively different from the single polarons merging presented in Ref. [23].

We agree with the Reviewer that it would be extremely interesting to study the energy competition between a singlet-bipolaron solution and the single polaron states. To date, a comprehensive analysis of the dynamical process bringing two single polarons together forming a bipolaron in real material remains unaddressed both at DFT and DMFT level, the reason being that this is an extremely difficult process to model, and BNOO does not offer the proper background to study this phenomenon. As the referee correctly points out, DMFT can provide insights on bipolaron formation. For example, DMFT applied to the Hubbard-Holstein model (Ref. [24]) shows that the formation of bipolaronic states at relatively large $U = 6t$ (for BNOO we have $U \approx 3t$) is possible. Transferring this analysis to a real materials would require a huge efforts, as a proper analysis would require DMFT calculations for a multiband Hubbard-Holstein model with realistic first-principles electron-phonon matrix elements. We plan to address this fundamental and challenging process in future studies.

Referee: As authors' state in the abstract spin-orbit and and polaron formation is "seemingly mutually exclusive". For a broad audience reader this statement it is not really obvious. Explanation indeed comes later in the discussion and it is far less trivial. I suggest the authors to take into account the literature on simple models discussing the interplay between electron-lattice and spin-orbit interaction. In simple models where the electron phonon-interaction is purely local (Holstein) a sort of competition was found between localization effects and spin-orbit by [Covaci et al.](<https://journals.aps.org/prl/abstract/10.1103/PhysRevLett.102.186403>). However this competition may turn into cooperation even for local electron-lattice interaction (see [Cappelluti et al.](<https://journals.aps.org/prb/abstract/10.1103/PhysRevB.76.085334>)) or Frölich (see [Grimaldi](<https://journals.aps.org/prb/abstract/10.1103/PhysRevB.81.075306>)).

Authors: We express our sincere gratitude to the Reviewer for bringing these studies to our attention, which we have found highly insightful. We have incorporated a comment along with the relevant references into the main text.

Referee: Fig. 2(a) hard to read symbols are too big. I suggest to split the figure and the inset in two in order to enlarge the y scale of the main panel and improve readability.

Authors: We have modified Fig. 2 according to the suggestion of the Reviewer.

Referee: IMT for Metal to Insulator Transition is non standard I suggest MIT.

Authors: We have adopted the standard formulation suggested by the Reviewer.

Bibliography

- ¹L. Lu, M. Song, W. Liu, A. P. Reyes, P. Kuhns, H. O. Lee, I. R. Fisher, and V. F. Mitrović, “Magnetism and local symmetry breaking in a Mott insulator with strong spin orbit interactions”, *Nature Communications* **8**, 14407 (2017).
- ²Y. Hibino, T. Taniguchi, K. Yakushiji, A. Fukushima, H. Kubota, and S. Yuasa, “Giant charge-to-spin conversion in ferromagnet via spin-orbit coupling”, *en*, *Nature Communications* **12**, 6254 (2021).
- ³H. Nakai and C. Hotta, “Perfect flat band with chirality and charge ordering out of strong spin-orbit interaction”, *en*, *Nature Communications* **13**, 579 (2022).
- ⁴H. Y. Huang, Z. Y. Chen, R.-P. Wang, F. M. F. de Groot, W. B. Wu, J. Okamoto, A. Chainani, A. Singh, Z.-Y. Li, J.-S. Zhou, H.-T. Jeng, G. Y. Guo, J.-G. Park, L. H. Tjeng, C. T. Chen, and D. J. Huang, “Jahn-Teller distortion driven magnetic polarons in magnetite”, *Nature Communications* **8**, 15929 (2017).
- ⁵C. Verdi, F. Caruso, and F. Giustino, “Origin of the crossover from polarons to fermi liquids in transition metal oxides”, *Nature Communications* **8**, 15769 (2017).
- ⁶M. Reticcioli, Z. Wang, M. Schmid, D. Wrana, L. A. Boatner, U. Diebold, M. Setvin, and C. Franchini, “Competing electronic states emerging on polar surfaces”, *en*, *Nature Communications* **13**, 4311 (2022).
- ⁷S. V. Streltsov and D. I. Khomskii, “Jahn-Teller Effect and Spin-Orbit Coupling: Friends or Foes?”, *Physical Review X* **10**, Publisher: American Physical Society, 031043 (2020).
- ⁸J. K. Kesavan, D. Fiore Mosca, S. Sanna, F. Borgatti, G. Schuck, P. M. Tran, P. M. Woodward, V. F. Mitrović, C. Franchini, and F. Boscherini, “Doping Evolution of the Local Electronic and Structural Properties of the Double Perovskite $\text{Ba}_2\text{Na}_{1-x}\text{Ca}_x\text{OsO}_6$ ”, *The Journal of Physical Chemistry C* **124**, 16577–16585 (2020).
- ⁹D. Fiore Mosca, L. V. Pourovskii, B. H. Kim, P. Liu, S. Sanna, F. Boscherini, S. Khmelevskiy, and C. Franchini, “Interplay between multipolar spin interactions, Jahn-Teller effect, and electronic correlation in a $J_{\text{eff}} = \frac{3}{2}$ insulator”, *Physical Review B* **103**, 104401 (2021).
- ¹⁰R. Cong, R. Nanguneri, B. Rubenstein, and V. F. Mitrović, “First principles calculations of the electric field gradient tensors of $\text{Ba}_2\text{NaOsO}_6$, a Mott insulator with strong spin orbit coupling”, *en*, *Journal of Physics: Condensed Matter* **32**, 405802 (2020).
- ¹¹A. Abragam, *The Principles of Nuclear Magnetism* (Clarendon Press, 1961).
- ¹²E. R. Andrew and D. P. Tunstall, “Spin-Lattice Relaxation in Imperfect Cubic Crystals and in Non-cubic Crystals”, *Proceedings of the Physical Society* **78**, 1 (1961).
- ¹³R. Cong, E. Garcia, P. C. Forino, A. Tasseti, G. Allodi, A. P. Reyes, P. M. Tran, P. M. Woodward, C. Franchini, S. Sanna, and V. F. Mitrović, “Effects of charge doping on Mott insulator with strong spin-orbit coupling, $\text{Ba}_2\text{Na}_{1-x}\text{Ca}_x\text{OsO}_6$ ”, *Physical Review Materials* **7**, 084409 (2023).
- ¹⁴D. C. Johnston, “Stretched exponential relaxation arising from a continuous sum of exponential decays”, *Physical Review B* **74**, Publisher: American Physical Society, 184430 (2006).
- ¹⁵J. C. Phillips, “Stretched exponential relaxation in molecular and electronic glasses”, *Reports on Progress in Physics* **59**, 1133 (1996).
- ¹⁶V. F. Mitrović, M.-H. Julien, C. de Vaulx, M. Horvatić, C. Berthier, T. Suzuki, and K. Yamada, “Similar glassy features in the ^{139}La nmr response of pure and disordered $\text{La}_{1.88}\text{Sr}_{0.12}\text{CuO}_4$ ”, *Phys. Rev. B* **78**, 014504 (2008).
- ¹⁷A. P. Dioguardi, T. Kissikov, C. H. Lin, K. R. Shirer, M. M. Lawson, H.-J. Grafe, J.-H. Chu, I. R. Fisher, R. M. Fernandes, and N. J. Curro, “Nmr evidence for inhomogeneous nematic fluctuations in $\text{BaFe}_2(\text{As}_{1-x}\text{P}_x)_2$ ”, *Phys. Rev. Lett.* **116**, 107202 (2016).

- ¹⁸P. M. Singer, A. Arsenault, T. Imai, and M. Fujita, “¹³⁹La Nmr investigation of the interplay between lattice, charge, and spin dynamics in the charge-ordered high- T_c cuprate $\text{La}_{1.875}\text{Ba}_{0.125}\text{CuO}_4$ ”, *Phys. Rev. B* **101**, 174508 (2020).
- ¹⁹H. Choi, I. Vinograd, C. Chaffey, and N. Curro, “Inverse laplace transformation analysis of stretched exponential relaxation”, *Journal of Magnetic Resonance* **331**, 107050 (2021).
- ²⁰C. Franchini, G. Kresse, and R. Podloucky, “Polaronic Hole Trapping in Doped BaBiO_3 ”, *Physical Review Letters* **102**, Publisher: American Physical Society, 256402 (2009).
- ²¹J. Tao and T. Liu, “Electron and Hole Polaron Formation and Transport in Monoclinic WO_3 Studied by Hybrid Functional Approach”, *The Journal of Physical Chemistry C* **127**, Publisher: American Chemical Society, 16204–16210 (2023).
- ²²K. E. Kweon, G. S. Hwang, J. Kim, S. Kim, and S. Kim, “Electron small polarons and their transport in bismuth vanadate: a first principles study”, en, *Physical Chemistry Chemical Physics* **17**, Publisher: The Royal Society of Chemistry, 256–260 (2014).
- ²³S. I. Emin, “Small polarons”, *Physics Today* **35**, 34 (1982).
- ²⁴P. Werner and A. J. Millis, “Efficient Dynamical Mean Field Simulation of the Holstein-Hubbard Model”, *Physical Review Letters* **99**, Publisher: American Physical Society, 146404 (2007).

REVIEWERS' COMMENTS

Reviewer #1 (Remarks to the Author):

The authors have made proper responses to my comments. Thus, I would like to recommend its publication in the current form.

I hope the authors will perform optical and transport measurements of the samples to obtain more direct evidence in the near future.

Reviewer #2 (Remarks to the Author):

The authors have adequately addressed all of the questions and comments, and the manuscript is improved. I recommend publication.

Reviewer #3 (Remarks to the Author):

Authors' rebuttal letter to all referees' comments reports convincing arguments.

In particular authors' have addressed my concern about competition of Hubbard repulsion and polaron formation as well as the objection of the second referee about the interpretation of the NMR data and modified the text and the supplementary information consistently.

The authors' reply to the comments of the first referee seems to me convincing. Both first and second referee expects some comments about transport in BNOO which as the authors comment are not present in literature up to now.

In conclusion I think the the paper should be published in Nature Communication.